# Invasive Pulmonary Aspergillosis: Not Only a Disease Affecting Immunosuppressed Patients

**DOI:** 10.3390/diagnostics13030440

**Published:** 2023-01-26

**Authors:** Rafael Zaragoza, Jordi Sole-Violan, Rachel Cusack, Alejandro Rodriguez, Luis Felipe Reyes, Ignacio Martin-Loeches

**Affiliations:** 1Critical Care Department, Hospital Universitario Dr. Peset, 46017 Valencia, Spain; 2Critical Care Department, Hospital Universitario Dr. Negrín, 35010 Las Palmas de Gran Canaria, Spain; 3Universidad Fernando Pessoa Canarias, 35010 Las Palmas de Gran Canaria, Spain; 4Department of Intensive Care Medicine, Multidisciplinary Intensive Care Research Organization (MICRO), St. James’s Hospital, D08 NHY1 Dublin, Ireland; 5Critical Care Department, Hospital Universitario Joan XXIII/URV/IISPV, 43005 Tarragona, Spain; 6Unisabana Center for Translational Science, Universidad de La Sabana, Chía 53753, Colombia; 7Department of Clinical Medicine, School of Medicine, Trinity College Dublin, D02 PN40 Dublin, Ireland; 8Hospital Clinic, Institut D’Investigacions Biomediques August Pi i Sunyer (IDIBAPS), Universidad de Barcelona, CIBERES, 08007 Barcelona, Spain

**Keywords:** IPA, CAPA, IAPA, *Aspergillus*, sepsis

## Abstract

Fungal infections have become a common threat in Intensive Care Units (ICU). The epidemiology of invasive fungal diseases (IFD) has been extensively studied in patients severely immunosuppressed over the last 20–30 years, however, the type of patients that have been admitted to hospitals in the last decade has made the healthcare system and ICU a different setting with more vulnerable hosts. Patients admitted to an ICU tend to have older age and higher severity of disease. Moreover, the number of patients being treated in ICU are often immunosuppressed as a result of the widespread use of immunomodulatory agents, such as corticosteroids, chemotherapy, and biological agents. The development of Invasive Pulmonary aspergillosis (IPA) reflects a different clinical trajectory to affected patients. The increasing use of corticosteroids would probably explain the higher incidence of IPA especially in critically ill patients. In refractory septic shock, severe community-acquired pneumonia (SCAP), and acute respiratory distress syndrome (ARDS), the use of corticosteroids has re-emerged in order to decrease unacceptably high mortality rates associated with these clinical conditions. It is also pertinent to note that different reports have used different diagnosis criteria, and this might explain the different incidence rates. Another layer of complexity to better understand current IPA data is related to more aggressive acquisition of samples through invasive respiratory examinations.

## 1. Introduction

Fungal infections have become a common thread in Intensive Care Units (ICU) [1,2]. The epidemiology of invasive fungal diseases (IFD) has been quite well studied in severely immunosuppressed patients over the last 20–30 years, however, the type of patient that has been admitted to ICU in the last decade has made the ICU a different setting with more vulnerable hosts [3]. Current ICU patients tend to be older and more severe, in addition to having acquired immunosuppression. This is due to the widespread use of immunomodulatory agents such as corticosteroids, chemotherapy, and biological agents [4,5,6].

There are primary distinctions regarding fungal infections that affect or do not affect the respiratory tract [7]. Infections of the respiratory tract are primarily caused by moulds and Aspergillus species. This manuscript outlines the development of an invasive pulmonary infection that consumes a great deal of medical resources consumption: Invasive Pulmonary Aspergillosis (IPA).

We will firstly explore the link with *Aspergillus* spp. and its pathophysiology. We will further explore the disease in three clinical conditions: IPA in immunosuppressed non neutropenic patients [8], influenza-associated pulmonary aspergillosis (IAPA) [9], and COVID-19-associated pulmonary aspergillosis (CAPA) [10].

## 2. Relevant Sections

### 2.1. Physiopathology

It is well known that the at-risk patient groups for developing IPA include immunocompromised, neutropenic patients, those receiving a solid organ transplant, individuals with human immunodeficiency virus (HIV), chronic obstructive pulmonary disease (COPD), liver failure, treatment with corticosteroids, and severe virus infections [11,12].

Colonization by *Aspergillus* spp. has also been described as a risk factor, particularly for patients with pulmonary structural damage [11]. *Aspergillus* fumigatus colonization of the airways in patients with COPD may progress to IPA in some cases. Moreover, fungal colonization of the airways in patients is associated with impaired lung function. The use of corticosteroids in this group of patients has also been described as a risk factor for the development of IPA [11,13].

In addition, *Aspergillus fumigatus* adds some other characteristics that can contribute to its pathogenicity, such as its very small spore size (3–5 mm), which enables them to penetrate deeply into the lung, several virulence determinants (proteases, ribotoxins, and other toxins), and protective substances that protect *Aspergillus* spp. from damage [12]. *Aspergillus* spp. conidia in the airway are properly cleared from the airway in a health host. However, in case of one or more defects in primary pulmonary immunity or secondary defences described previously, conidial germination into hyphal morphotypes occurs. This allows increased inflammatory responses in the airway and potential invasion into the lungs [14]. Following that, a mixed variety of inflammatory to invasive airway disease occurs, and invasive pulmonary aspergillosis can lead to different presentations depending on the risk factors involved, such as tracheobronchitis and post-obstructive pneumonia, and complications of invasive fungal pneumonia itself (nodules, necrosis with cavities, pleural invasion). *Aspergillus* spp. overgrowth in airways causes pathologic airway inflammation and excess mucous production. As a result, invasive aspergillosis tracheobronchitis leads to post-obstructive bacterial pneumonia, whereas at the alveolar level, hyphal growth causes invasive pneumonia [12,14].

Classically, two radiological and pathological patterns have been described (5). The first one-majority related with presence of severe neutropenia and/or severe bone marrow suppression consists of the development of one or more discrete nodules due to well-demarcated and round-shaped coagulation necrosis in which numerous hyphae aligned in a radial pattern. Generally, a circular band of haemorrhage surrounds the area of coagulation necrosis. Less apparent is any type of inflammatory infiltrate in this pattern. The second pattern, related to important response of neutrophils as a first defence line against *Aspergillus* spp. infection, is fused lobular consolidation (FLC), which corresponds to bronchopneumonia histologically characterized by the filling of acute inflammatory exudates with a fungal proliferation in alveoli. Necrosis present in FLC is usually colliquative and may be induced by neutrophilic infiltration. This can produce a cavity at the centre of a region when the bronchi involved in necrosis play a role in drainage [15].

The concept of severe viral infections (influenza or COVID-19) as predisposing factors for IPA is clear. Although both viral infections share some common risk factors and profound lymphopenia, the pathogenesis of both entities seems to be different. Extensive destruction of the respiratory epithelium by influenza favours the invasion of Aspergillus fumigatus by disruption of this natural barrier, which could explain the high proportion of tracheobronchitis. While SARS-CoV-2 enters alveolar epithelial cells and type II pneumocytes via the angiotensin-converting enzyme II (ACE II) receptors, its devastating effect is mainly related to endothelial cell damage, alteration of the renin-angiotensin system, and resulting ARDS, rather than destruction and necrosis of the respiratory tracheobronchial epithelium as observed in influenza. These facts could explain the angioinvasive form of IAPA in contrast with the extensive lung damage secondary to inflammation seen in CAPA [16,17]. Figure 1 illustrates the main features of IPA, as well as their clinical consequences.

### 2.2. Other Populations. Does It Really Exist as a Risk Factor in Non-Immunosuppressed?

It is widely described that *Aspergillus* spp. are ubiquitous in the environment with some ecological variations [18]. Aspergillosis is acquired by inhalation of *Aspergillus* spp. spores and colonizes the airways of patients [11]. There is a vast amount of literature published regarding IPA in severely immunocompromised patients, typically in neutropenia [19]. It is not surprising that local defences can become overwhelmed during the period of neutropenia [20]. A process called angioinvasion occurs, which results in significant destruction of the alveolar membrane [21]. In non-neutropenic hosts, spore inhalation rarely causes lung disease, however the number of cases in this subset of patients is rising. The most common risk factor identified has been the widespread use of corticosteroids [22].

Corticosteroids are powerful immunosuppressants that are commonly prescribed in certain clinical conditions, such as auto-immune disease and cancer, to modify the natural immune response. It is common for Intensive Care Medicine to prescribe corticosteroids in a very random manner. This can range from not being used after the publication of studies that show no benefits to using them very often as occurs today [23,24]. There may be a contagious effect for the use of corticosteroids in other medical conditions after their positive effects in patients with severe forms of COVID-19.

The following medical conditions have been shown, from time to time, to have IPA: patients with human immunodeficiency virus (HIV), COPD (both with oral or nebulised corticosteroid use), solid organ transplant recipients (particularly lung and heart-lung transplant recipients), and patients with liver failure [25]. An interesting feature in these patients, compared to those with neutropenia, is the lack of a “typical” radiological pattern that would include the halo sign that commonly occurs when angioinvasion is present [26].

Patients after lung transplants are an interesting group to consider, due to their high incidence and high mortality rates if the disease is present [27]. Risk factors include single lung transplantation, co-infection with cytomegalovirus, induction with alemtuzumab, and graft rejection with the use of monoclonal antibody [28]. In this type of patient, the use of bronchoscopy is more often implemented due to a high degree of suspicion. In IAPA and CAPA, it is often noted that ulcerations or pseudomembranes are present [29]. We must acknowledge, however, the increasing use of nebulised antifungals, especially azoles and polyenes. Recently, some studies have been conducted with a novel class of azole: opelconazole [30]. This azole differs from others, in that it is being evaluated and optimized for nebulized administration. We should see results within the next few years. These results will show if a decrease in IA would match the killing of spores in the airways of the transplanted host.

### 2.3. IAPA and CAPA Parallelisms and Differences

One of the most significant changes in the diagnosis and management of infectious diseases during the last decade has been the emergence of secondary infections (either as a co-infection or immediately after superinfection) in patients affected by community-acquired viral pneumonia [31,32,33,34,35]. The pathogenicity of viral infections in the lung has been extensively studied in animal and cellular models. There is no doubt that the damage associated with community acquired viral pneumonia due to influenza and COVID develops severe lung damage with an epithelial disruption that ultimately develops an acute respiratory distress syndrome (ARDS) with severe hypoxemia and, in some cases, with a fatal outcome due to multiorgan failure and refractory shock [36].

Some patients can be affected by co-infections that happen during the first days of hospital admission or immediately after. In the case of influenza, patients have been observed to have co-infection by Streptococcus pneumoniae in the majority of the cases. However, some large databases have also found the presence of other “non-common” pathogens such as Pseudomonas aeruginosa and, most surprisingly, *Aspergillus* spp. [34,37]. In the case of influenza, several authors reported unexpectedly high rates of the former. *Aspergillus* spp. Infections such as IPA have been mostly described in immunosuppressed patients with very few cases in immunocompetent patients, unless advanced COPD is reported as a patient’s comorbidity.

A key question is whether fungal acquisition and therefore infection could come from a nosocomial or endogenous source. Physiologically, IPA starts with alveolar damage, and then the fungus attaches to surfactant proteins, causing epithelial damage and angioinvasion. This is the primary reason why IPA is a deadly and difficult-to-treat disease, as it causes extensive lung tissue destruction in severe acute respiratory failure with severe forms of proliferative fibrosis like ARDS [21].

A classical concept has been that IPA occurs only in patients with severe immunosuppression. Many documents have been published including guidelines and taskforces that provide diagnosis and treatment recommendations always with the scope of immunosuppression [38,39,40]. This is conceptually something that has evolved over the years until patients with viral infections (influenza or COVID) started to develop unexpected events of clinical deterioration and the isolation of *Aspergillus* spp. from respiratory samples. The diagnosis always created some degree of confusion, as patients with viral co-infections with IPA did not show the typical patterns of halo or others [28]. Figure 2 shows the different radiological patterns between patients who are immunosuppressed or not with IPA.

There is still some confusion regarding the meaning of *Aspergillus* spp. isolation and its association with mortality in patients with viral pneumonia. During the COVID-19 pandemic, *Aspergillus* spp. isolated in respiratory samples of patients under invasive mechanical ventilation was widely reported. Some groups at the beginning of the pandemic reported unexpectedly high rates and high mortality rates [41,42]. The isolation of *Aspergillus* spp. does not manifest “per se” an IPA diagnosis [43]. This creates some difficulties when compared to previous non-COVID reports. Moreover, numerous definitions have been launched over the last few years, including one that aimed to be used in critically ill patients: putative [44]. Although this term has been well-implemented for research purposes, it is not widely used in clinical practice. In a multicentric observational study from France with almost 600 mechanically ventilated patients with COVID-19, it was found that the presence of CAPA was independently associated with a high mortality rate [45].

A recent paper by Rouze et al. analysed Blot et al.’s and Verweig et al.’s definition, and compared patients with COVID and Influenza and IPA to determine if IPA was more frequent in one or other entity, especially considering that in Influenza the use of corticosteroids is not advised, and in COVID-19, recent randomised clinical trials have found a survival benefit. Rouze et al. found that IPA was twice as frequent in patients with influenza than in COVID-19 [46]. This is a very surprising rate as, during COVID-19 times, there was a huge number of publications that reported a much higher incidence of CAPA [22]. In addition, some forms of IPA that have been previously reported in IAPA are more aggressive than others. Nyga et al. reported in an observational multicentric study that forms of invasive tracheobronchial aspergillosis have a much higher mortality at 30 days, 90 days, and ICU mortality [47]. Thus, there is a clear need for a more detailed examination of the lower tract respiratory airway to determine whether a lesion exists. This would not only provide valuable information but would also allow the patient to begin treatment as soon as possible.

In Figure 2, the differentiation from IAPA to CAPA is evident. Perhaps this is also related to overall radiology in these patients, as angio-invasion is common in patients with IAPA. Various microbiological laboratories use direct microscopy or calcofluor extensively for diagnosis [43]. This type of sampling can have some risks due to potential aerosol-generating procedures. There are other culture-based techniques that can be useful. However, there are several main pitfalls, including a high volume requirement, prolonged incubation, reduced diagnosis yields when concomitant antifungals are used, and in some cases it can be difficult to distinguish between colonization and infection. The “gold standard” for diagnosis is probably histology studies, but this diagnosis is often taken too late since it is difficult to obtain high quality samples. In Table 1, we discuss the potential reasons for the incidence of CAPA.

Finally, over the last years, several taskforces and guidelines have been proposed and published to have a clinical guidance to IPA management in both influenza and COVID-19 patients [22,29]. Among different guidelines, perhaps the common denominator has been the importance of prompt bronchoscopy diagnostic approach. This is to help to determine the inspection of large airways, to determine the presence of tracheal lesions, and to obtain bronchoalveolar lavage to process the determination of galactomannan in this fluid.

## 3. Discussion

Most guidelines are positioned to recommend the use of azoles as a first line agent. It is, however, important to consider some aspects with this valid recommendation [26]. The presence of azole resistant strains is a growing problem worldwide due to the increased use of fungicidal drugs in agriculture. Probably the most determinant problem is the potential problem of underdosing with some azoles such as voriconazole in critical care patients [48]. For this reason, the use of lipidic forms of amphotericin B should be considered, especially when azole resistance occurs in some local settings or voriconazole levels are not possible to determine [49].

The launch of new azoles might represent a step forward to optimise treatment success. Isavuconazole has been recommended by major guidelines. Isavuconazole belongs to the ‘triazole’ class of antifungal medicines. It works by disrupting the formation of ergosterol, an important component of fungal cell membranes. The main benefits of isavuconazole, mainly over voriconazole, but in some circumstances posaconazole as well, make it an alternative option for the management of complex patients with IPA. The main benefits are related to more predictable pharmacokinetics, a lack of QTc interval prolongation, improved tolerability, and a better profile for drug interaction when compared to voriconazole [50].

## 4. Conclusions

The development of IPA reflects a different clinical trajectory for affected patients. The increasing use of corticosteroids would probably explain the higher incidence of IPA especially in critically ill patients. In severe CAP (sCAP) and ARDS, the use of corticosteroids has re-emerged in order to decrease unacceptable high mortality rates associated with these clinical conditions. It is also pertinent to highlight that the different reports have used different diagnosis criteria, and this might explain the different incidence rates. Another layer of complexity to better understand current IPA data is related to more aggressive acquisition of samples through invasive respiratory examinations. The use of bronchoscopy has become widely used in critically ill patients over the last decade thanks to collaboration among different scientific societies and a multidisciplinary approach. This is to obtain the most accurate samples from severely ill patients.

## 5. Future Directions

The question now is what role CAPA will play in the event of future pandemics if they happen again. Based on the aforementioned features, CAPA is associated with a high mortality rate in patients who have recovered from ARDS. It is a secondary infection caused by immunosuppressive drugs. The features of CAPA differ from those seen in other IPA conditions and there is growing evidence that diagnosis is generally late. Should clinicians suspect CAPA more often? Should we start treatment sooner than when CAPA is confirmed? There are still many questions that need to be answered, but the most crucial message is to keep a high suspicion if the patient is developing a new episode of acute respiratory failure and sepsis. 

## Figures and Tables

**Figure 1 diagnostics-13-00440-f001:**
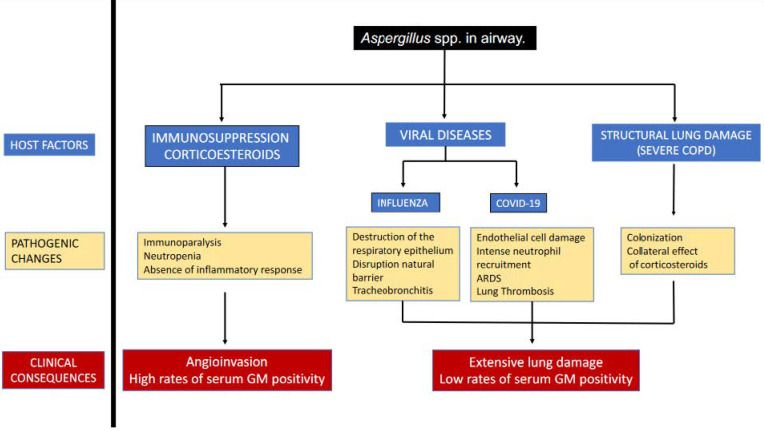
Pathophysiology and clinical consequences of IPA depending on host factors. Abbreviations: COPD: Chronic obstructive pulmonary disease, ARDS. Acute respiratory distress syndrome, GM: Galactomannan.

**Figure 2 diagnostics-13-00440-f002:**
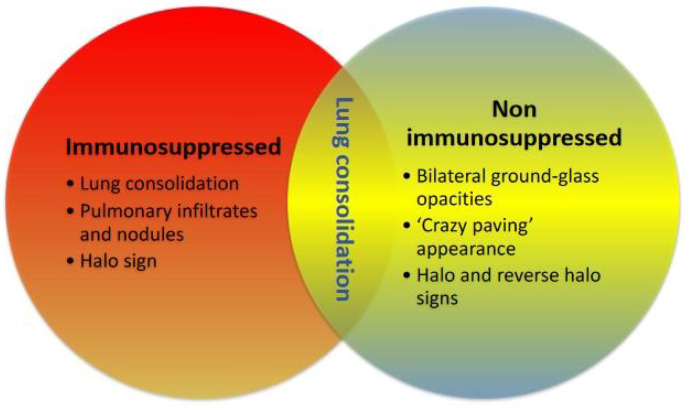
Radiological features of invasive pulmonary Aspergillosis if patients are or are not immunosuppressed.

**Table 1 diagnostics-13-00440-t001:** Potential reason of the heterogeneity of the reported incidence of patients with COVID-19-associated pulmonary aspergillosis (CAPA).

Various Definitions Used: EORTC/MSGERC; AspICU; ECMM/ISHAM
During the ‘first wave’ in early 2020 bronchoscopy was considered contraindicated in many centres, which probably led to under-reporting.
Criteria for defining CAPA relies on mycology, which may result in over-reporting.
Treatment modalities are evolving, which may impact on risk and incidence.

Abbreviations: EORTC/MSGERC: European Organization for the Research and Treatment of Cancer/Mycoses Study Group Education and Research Consortium. AspICU: A clinical algorithm to diagnose invasive pulmonary aspergillosis in critically ill patients. ECMM/ISHAM: European Confederation of Medical Mycology and the International Society for Human and Animal Mycology.

## Data Availability

The data presented in this study are available in article.

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
