# Peer review of "Invasive Pulmonary Aspergillosis: Not Only a Disease Affecting Immunosuppressed Patients"

_diagnostics, 2023, doi:10.3390/diagnostics13030440_

Round 1

Reviewer 1 Report

Thank you for giving the opportunity to review this article. I think the authors did a comprehensive review about the role of invasive pulmonary aspergillosis among immunocompetent patients. It provides updated information about this issue. Thus, I just had minor suggestion.

1. Please send for English editing for checking typo

2. May add more discussion about COVID-19 associated pulmonary aspergillosis.

Author Response

1/Please send for English editing for checking typo

We have a asked a native speaker to check the english

2/May add more discussion about COVID-19 associated pulmonary aspergillosis.

We have added a new paragraph in the expert opinion part regarding this comment.

Reviewer 2 Report

This manuscript provided a comprehensive review about this issue. I just have some minor suggestions.

1. Some typo, such as thread needs correction.

2. the anti-fungal treatment may be updated to mention the role of new azole

3. The format should be revised according to the journal's requirment.

Author Response

  1. Some typo, such as thread needs correction.

Response: The paper has been reviewed by a native speaker

  1. the anti-fungal treatment may be updated to mention the role of new azole

Response: We have added information of isavuconazole as requested

  1. The format should be revised according to the journal's requirment.

We have made changes in the format as requested following the structure: Abstract, Keywords, Introduction, Relevant Sections, Discussion, Conclusions, and Future Directions, 

Reviewer 3 Report

The article represents a good work, it shows how important it is to establish that invasive pulmonary aspergillosis is not only acquired by immunosuppressed patients, but also by other external causes such as corticosteroids that ultimately cause immunosuppression due to treatment of other diseases.  In this article it was observed that the number of words does not reach 4000, which is a requirement to be considered a reference article, therefore, I would suggest two alternatives, from which the authors could take 1

1. Expand the article by doing a systematic search and thus be considered a review article, or 2.

2. Leave the paper as a Perspective article

So I also suggest making some changes and corrections within the article.

L47 - spp. whitout italics.

L51 - spp. whitout italics.

L52-L54.  It is important to make reference to these three conditions in the abstract in order to obtain a better understanding of what the article is about in context.

L69, L70, L78, L89 - spp. whitout italics.

L109 - I suggest improving the quality of organization of the figure 1.

L162, L163, L179, L187, L189, L192- spp. whitout italics.

L199 - remove excess space.

L200 and L208 - Place period after et al.

L215 - remove excess space

Table 1. The font style and size of this box must be the same as that of the text.

Author Response

The article represents a good work, it shows how important it is to establish that invasive pulmonary aspergillosis is not only acquired by immunosuppressed patients, but also by other external causes such as corticosteroids that ultimately cause immunosuppression due to treatment of other diseases.  In this article it was observed that the number of words does not reach 4000, which is a requirement to be considered a reference article, therefore, I would suggest two alternatives, from which the authors could take 1

  1. Expand the article by doing a systematic search and thus be considered a review article, or 2.
  2. Leave the paper as a Perspective article

Response: We agree with the option of a perspective article.

So I also suggest making some changes and corrections within the article.

L47 - spp. whitout italics.

Response: Done

L51 - spp. whitout italics.

Response: Done

L52-L54.  It is important to make reference to these three conditions in the abstract in order to obtain a better understanding of what the article is about in context.

Response: Done

L69, L70, L78, L89 - spp. whitout italics.

Response: Done

L109 - I suggest improving the quality of organization of the figure 1.

Response: Done

L162, L163, L179, L187, L189, L192- spp. whitout italics.

Response: Done

L199 - remove excess space.

L200 and L208 - Place period after et al.

Response: Done

L215 - remove excess space

Response: Done

Table 1. The font style and size of this box must be the same as that of the text.

Response: Done

Round 2

Reviewer 3 Report

In the first review I requested that the letter in table 1 be the same as the letter in the text, so I request that the content be put back in table.  

Author Response

Sorry for this error, I have put in the same format Font Calibri Size 12